# Plasma pS129-α-Synuclein Is a Surrogate Biofluid Marker of Motor Severity and Progression in Parkinson’s Disease

**DOI:** 10.3390/jcm8101601

**Published:** 2019-10-03

**Authors:** Chin-Hsien Lin, Huei-Chun Liu, Shieh-Yueh Yang, Kai-Chien Yang, Chau-Chung Wu, Ming-Jang Chiu

**Affiliations:** 1Department of Neurology, National Taiwan University Hospital, College of Medicine, National Taiwan University, Taipei 100, Taiwan; q93421022@ntu.edu.tw; 2MagQu Co., Ltd., Xindian District, New Taipei City 231, Taiwan; gina.liu@magqu.com (H.-C.L.); syyang@magqu.com (S.-Y.Y.); 3MagQu LLC, 12425 W Bell Rd, C107, Surprise, AZ 85378, USA; 4Department of Internal Medicine, National Taiwan University Hospital, College of Medicine, National Taiwan University, Taipei 100, Taiwan; allenmy0920@gmail.com (K.-C.Y.); chauchungwu@ntu.edu.tw (C.-C.W.); 5Department of Pharmacology, College of Medicine, National Taiwan University, Taipei 100, Taiwan; 6Graduate Institute of Brain and Mind Sciences, College of Medicine, National Taiwan University, Taipei 100, Taiwan; 7Graduate Institute of Biomedical Engineering and Bioinformatics, National Taiwan University, Taipei 116, Taiwan; 8Graduate institute of Psychology, National Taiwan University, Taipei 116, Taiwan

**Keywords:** Parkinson’s disease, biomarker, α-synuclein, pS129-α-synuclein, motor severity

## Abstract

Phosphorylated α-synuclein accounts for more than 90% of α-synuclein found in Lewy bodies of Parkinson’s disease (PD). We aimed to examine whether plasma Ser129-phosphorylated α-synuclein (pS129-α-synuclein) is a surrogate marker of PD progression. This prospective study enrolled 170 participants (122 PD patients, 68 controls). We measured plasma levels of total and pS129-α-synuclein using immunomagnetic reduction-based immunoassay. PD patients received evaluations of motor and cognition at baseline and at a mean follow-up interval of three years. Changes in the Movement Disorder Society revision of the Unified Parkinson’s Disease Rating Scale motor score (MDS-UPDRS part III) and Mini-Mental State Examination (MMSE) score were used to assess motor and cognition progression. Our results showed that plasma levels of total and pS129-α-synuclein were significantly higher in PD patients than controls (total: 1302.3 ± 886.6 fg/mL vs. 77.8 ± 36.6 fg/mL, *p* < 0.001; pS129-α-synuclein: 12.9 ± 8.7 fg/mL vs. 0.8 ± 0.6 fg/mL, *p* < 0.001), as was the pS129-α-synuclein/total α-synuclein ratio (2.8 ± 1.1% vs. 1.1 ± 0.6%, *p* = 0.01). Among PD patients, pS129-α-synuclein levels were higher with advanced motor stage (*p* < 0.001) and correlated with MDS-UPDRS part III scores (*r* = 0.27, 95% CI: 0.09–0.43, *p* = 0.004). However, we found no remarkable difference between PD patients with and without dementia (*p* = 0.75). After a mean follow-up of 3.5 ± 2.1 years, PD patients with baseline pS129-α-synuclein > 8.5 fg/mL were at higher risk of motor symptom progression of at least 3 points in the MDS-UPDRS part III scores than those with pS129-α-synuclein < 8.5 fg/mL (*p* = 0.03, log rank test). In conclusion, our data suggest that plasma pS129-α-synuclein levels correlate with motor severity and progression, but not cognitive decline, in patients with PD.

## 1. Introduction

Parkinson’s disease (PD) is the second most common neurodegenerative disorder characterized by dopaminergic neuronal α-synuclein aggregations called Lewy bodies. The pathological α-synuclein can transmit from cell-to-cell in a prion-like fashion to promote the neurodegenerative process of this disease [1,2]. Although dopaminergic treatments could provide some symptomatic benefit for the motor symptoms of PD, their long-term use is problematic. As PD progresses, patients deteriorate in regard to both motor performance and cognitive function, and the expected mortality is two- to three-fold higher than that of the general population [3]. With the recent progress of mechanism-targeting therapies for PD, such as those targeting the propagation of α-synuclein using immunotherapeutic approaches into early human clinical trials [4], there is a pressing need for easily accessible molecular biomarkers that could track or predict disease progression in patients with PD.

Post-translational modifications of α-synuclein influence α-synuclein protein prone to misfolding and self-aggregation into fibrillar forms of species, and phosphorylated α-synuclein accounts for more than 90% of the α-synuclein found in Lewy bodies [5]. As most of the α-synuclein found in Lewy bodies is phosphorylated on serine 129 [6], it is possible that this modified, pathological form of the protein more accurately reflects the fundamental neuropathology of PD than total forms of α-synuclein [7]. Few studies focusing on cerebrospinal fluid (CSF) levels of phosphorylated α-synuclein as a diagnostic marker of PD have shown increased concentrations in patients with PD compared to controls and patients with atypical parkinsonism [8]. However, obtaining CSF is a relatively invasive procedure and is sometimes not amenable to every clinic, leading to the investigation of serum or plasma levels as alternatives. Although numerous studies have suggested that plasma levels of total α-synuclein are increased in patients with PD, the results are still conflicting [9,10,11]. We hypothesized that α-synuclein phosphorylated at Ser129 (pS129-α-synuclein) may better reflect disease-related alterations and be used as a surrogate marker for the severity and progression of PD. In this longitudinal follow-up study, we aimed to examine whether plasma levels of pS129-α-synuclein are associated with motor and cognitive progression in PD.

## 2. Experimental Section

### 2.1. Study Participants

Patients with PD were recruited from the movement disorder special clinics at a tertiary referral center in Taiwan. United Kingdom PD Society Brain Bank clinical diagnostic criteria was used to diagnose patients with PD [12]. Patients who had atypical or secondary parkinsonism syndromes were excluded. Healthy controls were recruited form the same institute. This study was reviewed by the institutional ethics committee of National Taiwan University Hospital and all participants provided informed consent before entering the study.

### 2.2. Evaluation of Motor and Cognitive Symptoms

This study had a prospective and longitudinal follow-up design. Patients with PD underwent evaluations of motor and cognitive function at baseline and a mean follow-up interval of 3 years. 

The severity of motor symptoms was evaluated using the motor subscale of the Movement Disorder Society revision of the Unified Parkinson’s Disease Rating Scale (MDS-UPDRS part III) [13]. Cognition was examined by the Mini-Mental State Examination (MMSE) [14]. PD with dementia (PDD) was diagnosed according to the diagnostic criteria suggested by the Movement Disorder Society (MDS) [15]. We used MMSE score ≤ 25 being the cut-off value for significant cognitive dysfunction in PD patients, as well as impairment of instrumental activities of daily living. Patients with severe dementia, defined as Clinical Dementia Rating (CDR) scale > 3 [16], were excluded to ensure that the recruited PDD patients can understand the instructions to complete the study. PD with mild cognitive impairment (PD-MCI) was diagnosed using the level I global cognitive function test suggested by the MDS diagnostic criteria with preservation of instrumental activities of daily living [17]. MMSE score of 26 to 28 was applied as a possible diagnostic feature, with an acceptable sensitivity and specificity rates of 0.92 and 0.42, for PD-MCI [18]. 

We assessed motor and cognitive progression based on changes in the MDS-UPDRS part III motor score and MMSE score during the follow-up period. Motor progression was defined as a sustained increase of at least 3 points in the MDS-UPDRS part III score at follow-up. Cognitive progression was defined as a sustained decrease in the MMSE score of at least 2 points during follow-up.

### 2.3. Measurement of Plasma Totalα-Synuclein and pS129-α-Synuclein

Ten milliliters of venous blood was drawn into tubes containing ethylenediamine tetracetic acid (EDTA) as anticoagulants from each participant at enrollment. Blood samples were centrifuged (2500× *g* for 15 min at 15–25 °C) within 60 min of collection and the plasma was stored at −80 °C for less than 3 months before testing. All of the aliquoted plasma samples were stored at −80 °C within 3 h after the blood was sampled before immunomagnetic reduction (IMR)-based immunoassay. Plasma levels of total α-synuclein were measured using the IMR α-synuclein kit (MF-ASC-0060, MagQu, Taipei, Taiwan) as described previously [19]. 

To assay plasma pS129-α-synuclein, we established the reagents for measuring pS129-α-synuclein in plasma: dextran-coating magnetic Fe_3_O_4_ nanoparticles (MF-DEX-0060, MagQu) bio-functionalized with monoclonal antibodies (825701, Biolegend) against phosphorylated Ser129 α-synuclein [20]. The mean diameter of the antibody-immobilizing magnetic nanoparticle was 56 nm, as detected by laser dynamic scattering (SZ100-S, HORIBA) [21]. The details of the methodologies for immobilizing antibodies against pS129-α-synuclein onto magnetic Fe_3_O_4_ nanoparticles, measuring the magnetic concentration of the immunocomplex, and the validation assay were described in the Appendix A.

### 2.4. Statistical Analysis

Numerical variables are expressed as mean ± standard deviation. Because the biomarker data compared in different groups in this study did not follow a Gaussian distribution and violated the assumptions of normality or homoscedasticity, the groups were compared by non-parametric Mann–Whitney *U* test (for two groups) or the Kruskal–Wallis test (for more than two groups). Correlations between variables were assessed by Pearson correlation analyses and the standardized correlation coefficients presented. The diagnostic accuracy of plasma pS129-α-synuclein levels was assessed by receiver operating characteristic curve (ROC) analyses. Youden’s *J* index was calculated for all points of the ROC curve, and the maximum value of the index was used as a criterion for selecting the optimal pS129-α-synuclein cut-off point for a diagnostic test with a numeric result. Kaplan–Meier curves were used to compare the cumulative probability risk of motor or cognition progression during the follow-up period between PD patients with pS129-α-synuclein levels above and below the cut-off value in the ROC analyses. Time zero for the Kaplan–Meier survival analysis was the date of plasma sample collection. For event-free patients, the follow-up was censored at the last clinic visit. The log rank test was used to determine any significant differences in the Kaplan–Meier curves between groups. We performed all analyses in Stata 8.0 (StataCorp LP, College Station, TX, USA) software. *p* < 0.05 was considered significant.

## 3. Results

### 3.1. Clinical Characteristics of Participants

A total of 190 study participants, including 122 PD patients and 68 healthy controls, were enrolled. The demographic and clinical information for all participants is summarized in Table 1. We found no significant differences in age, sex, or education between the two groups. The MMSE scores were lower in patients with PD than in controls (26.4 ± 2.3 vs. 29.3 ± 1.2, *p* < 0.01).

### 3.2. Cross-Sectional Analyses of pS129-α-Synuclein Levels and Motor and Cognitive Severity

We first examined whether age influences plasma total or pS129-α-synuclein concentrations in participants of various ages. The plasma concentrations of total or pS129-α-synuclein did not correlate with age in controls or patients with PD (total α-synuclein *r* = 0.12 (95% CI: −0.12–0.53), *p* = 0.20; pS129-α-synuclein: *r* = 0.05 (95% CI: −0.25–0.35), *p* = 0.73, Spearman correlation analysis), which is consistent with our previous study focusing on total α-synuclein [21]. 

The plasma levels of pS129-α-synuclein were significantly higher in PD patients (12.9 ± 8.7 fg/mL, minimal value: 0.1 fg/mL, maximal value: 67.4 fg/mL) than controls (0.8 ± 0.7 fg/mL, minimal value: 0.03 fg/mL, maximal value: 8.7 fg/mL; *p* < 0.001, Table 2, Figure 1A). The plasma levels of total α-synuclein and ratio of pS129-α-synuclein/total α-synuclein were also increased in PD patients compared to controls (total α-synuclein: PD vs. controls: 1302.3 ± 886.6 fg/mL vs. 76.4 ± 33.5 fg/mL, *p* < 0.001; pS129-α-synuclein/total α-synuclein ratio: PD vs. controls: 2.8 ± 1.1% vs.1.1 ± 0.6%, *p* = 0.01, Table 2, Figure 1B,C). The ROC analysis showed that a plasma pS129-α-synuclein concentration cut-off of 3.69 fg/mL had a sensitivity of 85.6% and specificity of 95.8% for distinguishing between PD and controls, with an area under the curve (AUC) of 0.94 (Figure 2). As total α-synuclein levels were also increased in patients with PD compared to controls (*p* < 0.001, Figure 1B), the plasma total α-synuclein cut-off of 119.5 fg/mL also had satisfactory sensitivity of 86.3% and specificity of 93.5% for distinguishing between patients with PD and controls (AUC 0.91; Figure 2). As both total and phosphorylated α-synuclein levels were increased in the PD group, the pS129-α-synuclein/total α-synuclein ratio cut-off of 0.22 had a 90.8% sensitivity and 58.3% specificity for distinguishing between patients with PD and controls (AUC 0.74; Figure 2).

Next, we examined whether plasma pS129-α-synuclein levels correlate with disease severity in terms of either motor or cognitive symptoms. In evaluating the severity of motor symptoms in PD, we found that pS129-α-synuclein levels increased as the motor symptom severity progressed (Table 3, Figure 3A). Specifically, pS129-α-synuclein levels were 11.1 ± 8.9 fg/mL for patients with early motor stage, defined as Hoehn–Yahr stages I and II, and 17.6 ± 10.2 fg/mL for patients with advanced motor stage, defined as Hoehn–Yahr stages III-V (*p* = 0.01). Furthermore, plasma pS129-α-synuclein concentrations in patients with PD correlated with motor symptom severity measured by the MDS-UPDRS part III scores (*r* = 0.27 (95% CI: 0.09–0.43), *p* = 0.004, Spearman correlation analysis, Figure 3B). However, in evaluating the relationship between plasma pS129-α-synuclein concentrations and various levels of cognitive ability among patients with PD, the plasma pS129-α-synuclein levels did not significantly differ between patients with varying severity of cognitive dysfunction (Figure 3C, *p* = 0.75). The pS129-α-synuclein levels were 13.7 ± 6.8 fg/mL, 11.5 ± 7.5 fg/mL, and 12.8 ± 7.1 fg/mL for the PD with normal cognition, PD-MCI, and PDD groups, respectively (*p* = 0.75). Plasma pS129-α-synuclein concentrations did not correlate with MMSE scores in patients with PD (*r* = 0.04 (95% CI: −0.14–0.22), *p* = 0.68). These findings suggest that higher pS129-α-synuclein levels are associated with more severe motor symptoms but not poorer cognitive performance in patients with PD. 

### 3.3. Longitudinal Follow-Up Analyses of Motor and Cognition Progression

After a mean follow-up of 3.5 ± 2.1 years, 36 of 122 patients (29.5%) with PD presented with a sustained increase of at least 3 points in the MDS-UPDRS part III scores. Baseline pS129-α-synuclein levels correlated with a decline in motor function. Cox regression analyses adjusted for age, sex, disease duration, and baseline MDS-UPDRS part III motor scores showed that higher baseline S129-α-synuclein levels were associated with a higher hazard ratio for motor symptom progression (adjusted hazard ratio 2.16 (0.96–4.91), *p* = 0.05). However, although a significant difference was found between the plasma levels of total form of α-synuclein between PD patients and controls (Figure 1B), the baseline total form of α-synuclein level was not associated with an increased risk of motor symptom progression after adjusting for age, sex, and baseline motor MDS-UPDRS part III scores (adjusted hazard ratio 1.08 (0.98–1.19), *p* = 0.09). These findings suggest that the plasma levels of phosphorylated α-synuclein could better associate motor symptoms progression than total α-synuclein. After calculating the sensitivity and specificity of this association, we performed a ROC analysis and identified cut-off values based on the highest Youden *J* index. We found that, among patients with PD, those with baseline pS129-α-synuclein levels > 8.46 fg/mL were at higher risk of motor symptom progression by at least 3 points in the MDS-UPDRS part III scores than those with pS129-α-synuclein levels < 8.46 fg/mL. Kaplan–Meier analysis showed a clear divergence between patient subgroups with pS129-α-synuclein levels above and below the cut-off value (Figure 4, *p* = 0.03, log rank test). 

On the other hand, during the follow-up period, 45 of 122 patients (36.8%) with PD presented with a persistent decline of at least 2 points in MMSE scores. A Cox regression analysis adjusted for age, sex, disease duration, and baseline MMSE scores showed that baseline pS129-α-synuclein levels do not associate with a higher hazard ratio for cognition progression (adjusted hazard ratios 0.99 (0.96–1.02), *p* = 0.63). 

## 4. Discussion

The results of this study demonstrate that plasma pS129-α-synuclein levels distinguish patients with PD and healthy controls and reflect motor symptom severity in patients with PD. We have, for the first time, demonstrated that plasma levels of pS129-α-synuclein correlate with MDS-UPDRS part III scores in PD patients. Furthermore, in a prospective follow-up of these patients with PD, we found that higher pS129-α-synuclein levels associate a higher risk of motor deterioration after considering the confounders age, sex, disease duration, and baseline motor severity score. 

Alpha-synuclein is the pathognomonic protein associated with PD. Mutation or multiplication of the α-synuclein gene (*SNCA*) causes familial forms of PD [22] and common variants of *SNCA* modulate the risk of sporadic PD [23]. Evidence have shown that α-synuclein could be released from neurons into body fluids, including CSF and plasma, contributing to cell-to-cell transmission of α-synuclein pathology in the disease process [1] and leading to numerous studies exploring plasma α-synuclein as a potential disease biomarker in patients with PD. However, plasma levels of α-synuclein are strongly influenced by red blood cell (RBC) contamination and hemolysis because RBCs are the major source (>99%) of α-synuclein in blood [24], contributing to the conflicting results for serum or plasma quantities of total α-synuclein in patients with PD compared to controls [9,10,11]. Approximately 90% of α-synuclein deposited in Lewy bodies is phosphorylated at Ser129, whereas ≤4% of total α-synuclein is phosphorylated at this residue in the normal brain [5,6], suggesting that Ser129-phosphorylated α-synuclein is one of the key players leading to the formation of Lewy bodies and may contribute to dopaminergic neurodegeneration in PD. In order to avoid some of the confounding issues caused by RBCs lysis in plasma, one previous study has attempted to specifically measure the pathogenic pS129-α-synuclein in plasma samples from patients with PD compared to healthy controls [25]. According to this case-control study [25], plasma pS129-α-synuclein is higher in PD patients than controls, with an AUC for diagnostic accuracy of 0.71. Our findings are in line with this study [25], in that we also observed a significantly higher plasma level of pS129-α-synuclein in PD patients compared to healthy controls, with an AUC for diagnostic accuracy of 0.92. One of the possible reasons for our better diagnostic accuracy may be due to the assay we applied to detect pS129-α-synuclein. Most of the studies exploring CSF or plasma biomarkers mainly utilized enzyme-linked immunosorbent assays (ELISAs) or similar immunoassays [25], which are often performed manually and are difficult to standardize, resulting in substantial variability in measurements between laboratories [26]. Using an ultra-sensitive IMR assay [19,20,21], which could quantitatively detect pS129-α-synuclein at ultra-low concentrations in plasma, our findings support the potential utility of plasma phosphorylated α-synuclein as a disease marker for PD.

Whether phosphorylated α-synuclein in plasma originates mainly from a peripheral tissue source or from the brain is unclear. The level of cytosolic, soluble, non-phosphorylated α-synuclein decreases in vulnerable brain regions during the disease process of PD, and α-synuclein becomes increasingly more phosphorylated and insoluble as the disease progresses. The percentage of phosphorylated α-synuclein increased from a normal level of approximately 5% to a final level of 30–100% depending on the brain regions and disease severity in PD [27]. Therefore, the significantly increased pS129-α-synuclein in plasma may likely originate from the brain via leakage or secretion across the blood–brain barrier. On the other hand, according to the Braak pathology staging hypothesis, pS129-α-synuclein with Lewy body pathology starts in the enteric nervous system of gastrointestinal tract, and then spreads to the medulla oblongata and pontine tegmentum, followed by the substantia nigra and other areas of the midbrain and basal forebrain as the disease progresses to its symptomatic stage [28]. Changes in the pathological pS129-α-synuclein in the blood plasma could reflect this temporal pattern of pathological change and, if they have a peripheral origin, could provide a marker for early-stage detection of the disease. 

Our study also showed that plasma pS129-α-synuclein levels correlate with MDS-UPDRS part III motor scores in PD patients. Furthermore, higher levels of plasma pS129-α-synuclein were associated with faster motor symptom progression in PD patients during the follow-up period, even after considering confounder effects, including age, sex, and baseline motor symptom severity. In addition to neuronal Lewy body accumulations, progressive neurodegeneration of dopaminergic neurons in the substantia nigra is the pathological substrate of classical motor symptoms in PD. Studies have shown that motor symptoms occur in PD patients when there is ≥ 70% loss of dopaminergic neurons in the substantia nigra [29,30]. A negative correlation was observed between dopaminergic neuron density and local α-synuclein burden in the nigral neurons of PD patients (ρ  =  −0.54), suggesting the severity of neurodegeneration and local burden of α-synuclein pathological conditions are closely coupled during disease progression in PD [31]. One recent large autopsy study showed that nigral neuron loss and phospho-α-synuclein pathological loadings are associated with the severity of signs of parkinsonism in the elderly [32]. These observations reinforce our findings that plasma pS129-α-synuclein may serve as a surrogate marker of motor symptom severity and progression in PD. In contrast, we did not find a correlation between plasma levels of pS129-α-synuclein and MMSE scores or the severity of cognitive dysfunction in patients with PD. These data suggest that plasma pS129-α-synuclein alone is not sufficient to determine the risk of dementia in PD patients. Cortical Lewy body pathology is more extensive in PDD than in PD without dementia, suggesting that the plasma pS129-α-synuclein burden is more severe in PDD than in PD with normal cognition. However, amyloid β plaques and tau neurofibrillary tangles, the hallmark pathologies of Alzheimer’s dementia, are also observed and correlate with cognitive status in patients with PDD [33]; thus, future longitudinal follow-up studies concomitantly incorporating an assessment of pS129-α-synuclein, α-synuclein, amyloid β protein, and total and phospho-tau in plasma to predict the risk of PDD are needed. 

The major advantage of this study was enrolling PD patients with varying disease severity, as well as control participants, in a prospective follow-up study design. However, our study has some limitations. First, the number of normal controls and patients with PD was not comparable. However, the sample size in our study reaches significant statistic power (>90%) while considering the mean value of pS129-α-synuclein in each group and setting the type I error as 0.05 [33]. Second, we only assessed cognitive function using the MMSE, a simple measurement of global cognitive function. Detailed neuropsychological tests evaluating individual cognitive domains are warranted for further assessment of the correlation between plasma pS129-α-synuclein level and individual cognitive decline in PD patients. Third, other post-translational modified forms of α-synuclein, including nitrated, glycated, and SUMOylated α-synuclein, which are also observed in the Lewy bodies, were not assayed in the current study. We also did not check the plasma levels of amyloid β and tau in this study. As amyloid β plaques and tau neurofibrillary tangles are also observed post-mortem in patients with advanced PD [34], future longitudinal follow-up studies concomitantly incorporating assessments of different post-translationally modified forms of α-synuclein, amyloid β protein, and total and phospho-tau in plasma may be needed to better predict PD progression. Finally, we did not have the CSF samples of all participants in our study, which hampered our ability to examine the phospho-S129-α-synuclein level in the CSF or the measurement of CSF/serum albumin ratio for checking the integrity of blood–brain barriers. Future studies that examine phospho-S129-α-synuclein levels in both CSF and plasma samples are warranted to further understand the potential role of phospho-S129-α-synuclein as being a biomarker for PD progression.

In summary, our findings suggest that plasma pS129-α-synuclein may serve as a surrogate biomarker of disease severity and progression in PD in regard to motor function, and our results may have potential for monitoring responses to therapy in future mechanism-targeted therapeutic trials.

## Figures and Tables

**Figure 1 jcm-08-01601-f001:**
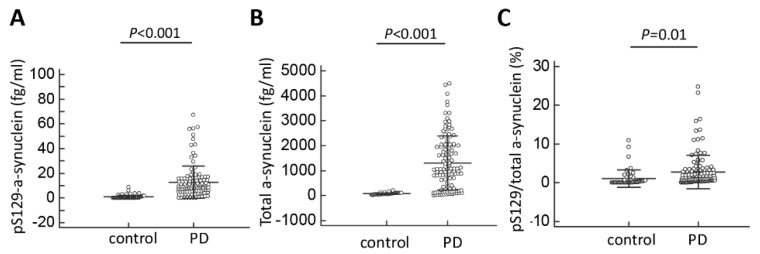
Plasma total and phosphorylated Ser129-α-synuclein levels for all participants in the study. (**A**) The plasma levels of pS129-α-synuclein, (**B**) total α-synuclein level and (**C**) the pS129/total α-synuclein ratio were significantly increased in patients with Parkinson’s disease (PD) compared to normal controls.

**Figure 2 jcm-08-01601-f002:**
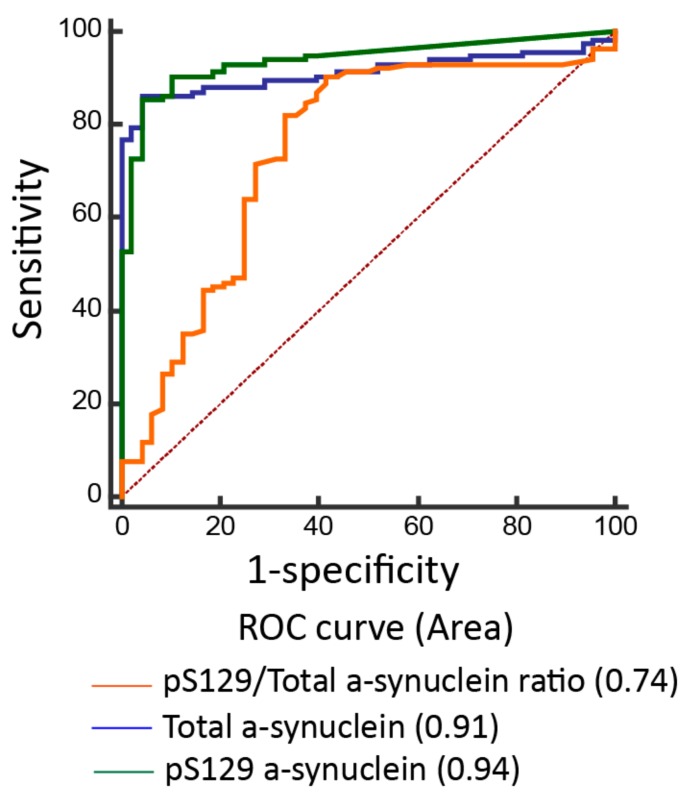
Receiver operating characteristic curves for predicting disease in participants. The accuracy of predicting Parkinson’s disease using the pS129/total α-synuclein ratio (area under curve (AUC) = 0.63), total α-synuclein level (AUC = 0.91), and pS129-α-synuclein (AUC = 0.94).

**Figure 3 jcm-08-01601-f003:**
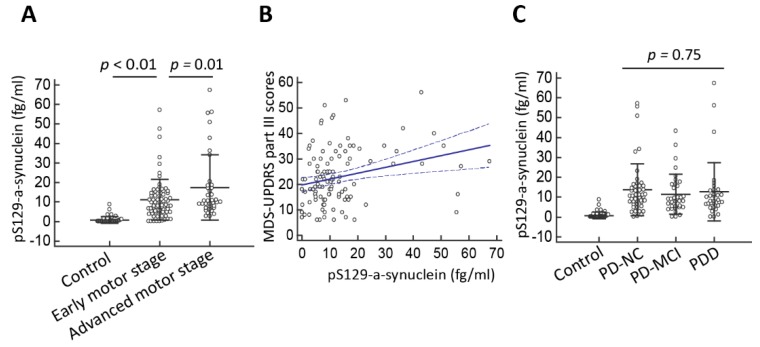
Plasma pS129-α-synuclein levels in Parkinson’s disease (PD) patients with varying disease severity. (**A**) The plasma pS129-α-synuclein level was markedly increased in PD patients with more severe motor disability as assessed by Hoehn–Yahr stage (*p* < 0.01) and (**B**) correlated with motor symptom severity as measured by MDS-UPDRS part III scores (*r* = 0.27 (95% CI: 0.09–0.43), *p* = 0.004). (**C**) The plasma pS129-α-synuclein levels did not differ between patients with varying severity of cognitive dysfunction (*p* = 0.75). Numbers are expressed as mean ± standard deviation. PD-NC, PD with normal cognition; PD-MCI, PD with mild cognitive impairment; PDD, PD with dementia.

**Figure 4 jcm-08-01601-f004:**
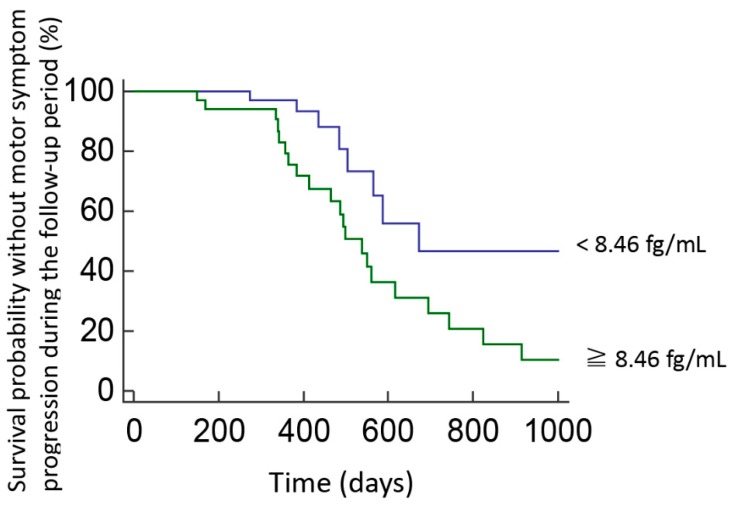
Survival probability without motor symptom progression in PD patients with high or low plasma pS129-α-synuclein levels during follow-up. Kaplan–Meier plots show survival probability without motor progression in patients with PD who had baseline pS129-α-synuclein concentrations above or below the cut-off levels determined in ROC analyses. Motor progression was defined as a sustained increase of at least 3 points in the MDS-UPDRS part III score at follow-up.

**Table 1 jcm-08-01601-t001:** Clinical characteristics and plasma biomarker levels of PD patients and controls.

Characteristics	Controls (*n* = 68)	PD (*n* = 122)	*p* Value
Age, years	68.3 ± 9.3	69.3 ± 10.1	0.32
Sex, male	44.1%	46.7%	0.72
Education, years	12.9 ± 4.2	11.7 ± 3.8	0.72
Disease duration, years	N.A.	6.9 ± 3.7	N.A.
MMSE	29.3 ± 1.2	26.4 ± 2.3	<0.01
Hoehn–Yahr stages, on	N.A.	2.2 ± 0.9	N.A.
Hoehn–Yahr stages, off	N.A.	3.2 ± 1.8	N.A.
MDS-UPDRS part III scores, on	N.A.	16.8 ± 8.3	N.A.
MDS-UPDRS part III scores, off	N.A.	32.2 ± 10.8	N.A.

PD: Parkinson’s disease; MMSE: Mini-Mental State Examination; N.A.: not available; MDS-UPDRS: Movement Disorder Society version of Unified Parkinson’s Disease Rating Scale.

**Table 2 jcm-08-01601-t002:** Plasma biomarker levels of study participants.

	Normal Controls (*n* = 68)	PD Patients (*n* = 122)	*p* Value
Total α-synuclein, fg/mL (minimal–maximal values)	76.4 ± 33.5 (21.0–211.5)	1302.3 ± 886.6 (5.7–4477.9)	*p* < 0.01 **
pS129-α-synuclein, fg/mL (minimal–maximal values)	0.8 ± 0.7 (0.03–8.7)	12.9 ± 8.7 (0.1–67.4)	*p* < 0.01 **
pS129-α-synuclein/total α-synuclein ratio	1.1 ± 0.6% (0.01–10.8)	2.8 ± 1.1% (0.01–24.8)	*p* = 0.01 *

PD, Parkinson’s disease. Numbers are expressed as mean ± standard deviation. * *p* < 0.05; ** *p* < 0.01. *p*-values were obtained from comparisons of individual characteristics between individual groups of participants using analysis of variance (ANOVA). For variables that did not display a normal distribution, data were compared with the Kruskal–Wallis test, the non-parametric equivalent of the independent sample *t*-test.

**Table 3 jcm-08-01601-t003:** Clinical characteristics and plasma biomarker levels of PD patients with different severity.

	PD (*n* = 122)		*p* Value		PD (*n* = 122)		*p* Value
	Early Motor Stage (*n* = 76)	Advanced Motor Stage (*n* = 46)		PD-NC (*n* = 51)	PD-MCI (*n* = 35)	PDD (*n* = 36)	
Age (years)	67.4 ± 10.2	73.1 ± 8.7	*p* = 0.02	64.3 ± 10.9	70.3 ± 6.4	79.9 ± 8.3	*p* < 0.01 **
Gender (M, %)	46.1%	50.0%	*p* = 0.12	47.1%	48.5%	50.3%	*p* = 0.21
Disease duration (years)	4.9 ± 3.5	8.0 ± 4.6	*p* = 0.02	4.9 ± 2.6	6.1 ± 4.2	6.7 ± 3.3	*p* = 0.04 *
MMSE	26.8 ± 3.1	22.7 ± 3.6	*p* = 0.03	28.8 ± 0.8	26.3 ± 0.9	18.8 ± 4.5	*p* < 0.01 **
Hoehn–Yahr stage (on)	1.5 ± 0.5	3.0 ± 0.9	*P* < 0.01 **	1.5 ± 0.9	2.1 ± 0.7	2.7 ± 1.3	*p* = 0.06
Hoehn–Yahr stage (off)	1.9 ± 0.9	3.3 ± 1.2	*p* = 0.02	1.8 ± 1.3	2.5 ± 0.9	3.2 ± 1.2	*p* = 0.07
UPDRS part III (on)	19.1 ± 8.5	30.5 ± 12.1	*p* < 0.01 *	18.8 ± 8.5	23.6 ± 9.7	27.6 ± 10.2	*p* = 0.04 *
UPDRS part III (off)	28.8 ± 10.8	42.6 ± 13.4	*p* < 0.01 *	30.0 ± 7.4	32.5 ± 6.2	37.1 ± 14.3	*p* = 0.09
Total α–synuclein, fg/mL (minimal–maximal values)	1322.8 ± 1136.6 (19.5–4427.8)	1423.2 ± 1023.8 (32.7–4477.9)	*p* = 0.57	1181.7 ± 1115.6 (34.8–4477.9)	1417.7 ± 1174.6 (32.7–4427.8)	1371.6 ± 929.4 (5.6–3762.4)	*p* = 0.68
pS129–α–synuclein, fg/mL (minimal–maximal values)	11.1 ± 8.9 (0.1–56.2)	17.6 ± 10.2 (2.8–67.4)	*p* = 0.01 *	13.7 ± 6.8 (0.2–56.2)	11.5 ± 7.5 (0.1–43.3)	12.8 ± 7.1 (0.1–67.4)	*p* = 0.75
pS129–α–synuclein/total α–synuclein ratio	2.3 ± 1.5% (0.01–16.2)	3.6 ± 1.9% (0.02–21.2)	*p* = 0.05	2.9 ± 1.6% (0.01–21.2)	2.2 ± 1.4% (0.01–16.2)	2.7 ± 1.8% (0.02–17.4)	*p* = 0.90

PD, Parkinson’s disease; MCI, mild cognitive impairment; PDD, Parkinson’s disease dementia; MMSE, mini-mental status examination; UPDRS, Unified Parkinson’s Disease Rating Scale; Numbers are expressed as mean ± standard deviation. * *p* < 0.05; ** *p* < 0.01. *p*-values were obtained from comparisons of individual characteristics between individual groups of participants using analysis of variance (ANOVA). For variables that did not display a normal distribution, data were compared with the Kruskal–Wallis test, the non-parametric equivalent of the independent sample *t*-test.

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
