# Peer review of "Plasma pS129-α-Synuclein Is a Surrogate Biofluid Marker of Motor Severity and Progression in Parkinson’s Disease"

_jcm, 2019, doi:10.3390/jcm8101601_

Round 1
Reviewer 1 Report
The manuscript “Plasma pS129-a-synuclein is a surrogate biofluid marker of motor severity and progression in Parkinson's disease” is an interesting paper, which can be published in the current form.
Author Response
Reviewer 1: The manuscript “Plasma pS129-a-synuclein is a surrogate biofluid marker of motor severity and progression in Parkinson's disease” is an interesting paper, which can be published in the current form.
Reply: We thank the reviewer for the kind words.
Reviewer 2 Report
In this research article, the authors investigated the level of total and phosphorylated alpha-synuclein (pS129-a-synuclein) in plasma of patients affected by Parkinson Disease (PD) and healthy subjects using immunomagnetic reduction-based immunoassay. In a follow-up prospective, they also evaluated PD patients’ motor and cognitive functions at baseline and after about 3.5 years. Both total and pS129-a-synuclein levels are significantly higher in PD group compared to control group. Focusing on pS129-a-synuclein, they found higher values in PD patients with advanced motor stage, suggesting that plasma pS129-a-synuclein correlates with motor severity. Finally, analyzing motor and cognitive progression after 3.5 years, the authors say that baseline pS129-a-synuclein levels correlate with a decline in motor functions. Based on these data, they propose plasma pS129-a-synuclein as a surrogate marker for motor severity and progression in PD.
This is an interestingly study focused on the identification of a reliable marker for PD progression. Howevere, many parts should be ameliorated and some questions discussed.
1- The main point that could weaken the obtained results is about the different size of the two compared groups (healthy subjects n=48 vs PD patients n=122). Why only 48 healthy controls have been investigated? To strongly validate the obtained results, it is necessary to recruit additional healthy subjects.
2- The introduction should be improved to better introduce the scientific background of this work. Furthermore, some sentences should be sustained by properly references. For example, why the authors say that CSF is not amenable to a longitudinal study (pag2 lines 62)? Is this sentence based on some data? If so, references are needed. However, CSF has been used in many longitudinal studies also to detect possible PD biomarkers (see for example Mollenhauer B et al., 2019)
3- Pag4: Table 1 is not clear. I suggest to split it into two tables, one for the demographic and clinical characteristics of the investigated subjects, another one for their plasma biomarker levels. Furthermore, in addition to mean and standard deviation, min and max values should be reported.
4- Pag5: Descriptions of panel A and B are wrong in the figure legend.
5- Pag6: This figure is not clear. First, we don't know the composition of the different PD- subgroups, since these data are not indicated by the authors. Among the 122 PD patients, how many are characterized by early or advanced motor stage? and how many show normal cognition, mild cognitive impairment or dementia? I didn't find these data in the manuscript. Second, all p values should be clearly indicated in the panels. Third, I understood that the values showed in this figure are the same previously showed in figure 1. If it is right, why the higher values showed in Figure 3 are about 70 fg/ml while in Figure 1 higher values are also present? For the statistical analysis, these values has been considered or omitted?
6- Pag7 Figure 4. What is showed in Figure 4: motor symptom progression or the survival Probabily? What has been evaluated by the authors?
7- pag8 line 24 "ser129-phosphorilated a-synuclein is the key executor leading to the formation of Lewy bodies and dopaminergic neurodegeneration in PD". Are the authors sure about this?
8- pag 8 line247-252. Which are these "few studies"? It is not clear where the authors observed a significant higher plasma level: in [25] or in their work?
Author Response
Reviewer 2: In this research article, the authors investigated the level of total and phosphorylated alpha-synuclein (pS129-a-synuclein) in plasma of patients affected by Parkinson Disease (PD) and healthy subjects using immunomagnetic reduction-based immunoassay. In a follow-up prospective, they also evaluated PD patients’ motor and cognitive functions at baseline and after about 3.5 years. Both total and pS129-a-synuclein levels are significantly higher in PD group compared to control group. Focusing on pS129-a-synuclein, they found higher values in PD patients with advanced motor stage, suggesting that plasma pS129-a-synuclein correlates with motor severity. Finally, analyzing motor and cognitive progression after 3.5 years, the authors say that baseline pS129-a-synuclein levels correlate with a decline in motor functions. Based on these data, they propose plasma pS129-a-synuclein as a surrogate marker for motor severity and progression in PD.
This is an interestingly study focused on the identification of a reliable marker for PD progression. However, many parts should be ameliorated and some questions discussed.
Question 1: The main point that could weaken the obtained results is about the different size of the two compared groups (healthy subjects n=48 vs PD patients n=122). Why only 48 healthy controls have been investigated? To strongly validate the obtained results, it is necessary to recruit additional healthy subjects.
Reply: We thank the reviewer for the comments. We have added additional 20 control samples in this revision with a final total number of 68 controls and 122 PD patients. The sample size in our study now reaches significant statistic power (>90%) while considering the mean value of pS129-a-synuclein in each group and setting the type I error as 0.05 (Rosner B. Fundamentals of Biostatistics. 7th ed. Boston, MA: Brooks/Cole; 2011).
We have listed this point as one of our study limitations in the “Discussion” section, in line 322 on page 9; and lines 323-324 on page 10.
Question 2: The introduction should be improved to better introduce the scientific background of this work. Furthermore, some sentences should be sustained by properly references. For example, why the authors say that CSF is not amenable to a longitudinal study (pag2 lines 62)? Is this sentence based on some data? If so, references are needed. However, CSF has been used in many longitudinal studies also to detect possible PD biomarkers (see for example Mollenhauer B et al., 2019)
Reply: We thank the reviewer for the comments. We agree with the reviewer that CSF has been used in many longitudinal studies in medical centers. However, obtaining CSF is a relatively invasive procedure and is sometimes not amenable to every clinic, leading to the investigation of serum or plasma levels as alternatives.
We have amended this sentence in the “Introduction” section, in lines 61-63, on page 2.
Question 3: Pag4: Table 1 is not clear. I suggest to split it into two tables, one for the demographic and clinical characteristics of the investigated subjects, another one for their plasma biomarker levels. Furthermore, in addition to mean and standard deviation, min and max values should be reported.
Reply: We thank the reviewer for this kind and helpful comment. As the reviewer suggested, we have split the original table 1 into two tables. The revised table 1 showed the clinical characteristics between controls and PD patients. The newly added table 2 showed the plasma biomarker data, including the minimal and maximal values, in each group on page 4.
Question 4: Descriptions of panel A and B are wrong in the figure legend.
Reply: We thank the reviewer for the correction. We have corrected and switched the panel A and B in the figure legend of Figure 1.
Please refer to line 173 on page 5.
Question 5: This figure is not clear. First, we don't know the composition of the different PD- subgroups, since these data are not indicated by the authors. Among the 122 PD patients, how many are characterized by early or advanced motor stage? and how many show normal cognition, mild cognitive impairment or dementia? I didn't find these data in the manuscript. Second, all p values should be clearly indicated in the panels. Third, I understood that the values showed in this figure are the same previously showed in figure 1. If it is right, why the higher values showed in Figure 3 are about 70 fg/ml while in Figure 1 higher values are also present? For the statistical analysis, these values has been considered or omitted?
Reply: We thank the reviewer for the comments and suggestions. To better know the compositions of the different PD subgroups, we have added a new table 3 that illustrated the clinical characteristics and plasma biomarker levels in PD patients with different motor or cognitive severities. We hope that this new table can make us clearer. Please kindly refer to
page 6 for the Table 3.
For the figure, we are sincerely grateful to the reviewer’s comments. The Figure 1A was replaced by a revised new one for the reasons that we added 20 new control samples and we actually have replaced the 5 outliers that the data was beyond 3 standard deviations of the mean value by a repeated double-check data. The data used for the figure 2 and 3 are actually the newly double-checked data of all PD patients. Therefore, the maximal value of pS129- a -synuclein is 67.4 fg/ml. We have detailed the data in the newly added table 3 on page 6 and amended Figure 1A on page 5 accordingly. We sincerely thank the reviewer for the correction once again.
Question 6: Figure 4. What is showed in Figure 4: motor symptom progression or the survival Probabily? What has been evaluated by the authors?
Reply: We thank the reviewer for the comments. The Figure 4 showed “Survival probability without motor symptom progression during the follow-up period (%)”. We have reworded the subtitle of Y-axis of Figure 4 on page 8 to make us clearer. Motor progression was defined as a sustained increase of at least 3 points in the MDS-UPDRS part III score at follow-up. This definition was addressed in the “Experimental Section”, in lines 97-98, on page 3; and was added in the figure legend for revised Figure 4 on page 8.
Question 7: pag8 line 24 "ser129-phosphorilated a-synuclein is the key executor leading to the formation of Lewy bodies and dopaminergic neurodegeneration in PD". Are the authors sure about this?
Reply: We thank the reviewer for the comments and suggestions. We have amended and lowered the tone of this sentence based on the current scientific evidence that described in the “Discussion” section. The amended sentence is as below.
Ser129-phosphorylated a-synuclein is one of the key players leading to the formation of Lewy bodies and may contribute to dopaminergic neurodegeneration in PD.
This point has been addressed in the “Discussion” section, in lines 266-267, on page 8.
Question 8: pag 8 line247-252. Which are these "few studies"? It is not clear where the authors observed a significant higher plasma level: in [25] or in their work?
Reply: We thank the reviewer for the comments. There is only one previous study exploring plasma pS129-a-synuclein level in a case-control study, which was cited as reference 25. We have amended the sentence as below to avoid readers’ confusion.
One previous study has attempted to specifically measure the pathogenic pS129-a-synuclein in plasma samples from patients with PD compared to healthy controls [25].
This point has been addressed in the “Discussion” section, in lines 268-270, on page 8.
Reviewer 3 Report
In this manuscript the authors investigate if the plasma phosphorylated form of alpha-synuclein is a surrogate marker of PD progression. The measurements of pS129-alpha-synuclein were performed using the ultra-high-sensitivity detection technology for protein immunomagnetic reduction (IMR) immunoassay. I have the following comments:
1. Please provide the anticoagulant used during collection of blood samples.
2. Clarifications of the statistics used in the manuscripts are needed. Please do not alternate methods for Gausssian distributed variables and non-parametric tests. It would be easier to follow if only non-parametric tests are applied, expressing the values and levels as medians (inter-quartile ranges), Spearman correlations, Mann-Whitney, ANCOVA etc.
An alternative, is to provide the statistic method used in each figure, table, results etc. Also providing the evidence for the values that had a Gaussian distribution.
There seems be some patients that have extremely high levels of pS129-alpha-synuclein. Have the authors checked if these values could be regarded as ‘outliers’?
3. The number of patients with PD (n=122) were many more than the healthy controls (n=48). Was there a certain study design (including the number patients and healthy controls) prior to the analyses of the plasma samples and statistical evaluation of the pS129-alpha-synuclein?
4. As the authors comment on, we can only speculate regarding the origin of alpha-synclein released into blood. In a recent study by Cariulo et al, Phospho-S129 Alpha-Synuclein Is Present in Human Plasma but Not in Cerebrospinal Fluid as Determined by an Ultrasensitive Immunoassay, they was not able to measure pS129P-alpha-synuclein in CSF samples using an in-house assay based on the Single-molecule counting technology (Singulex Erenna Immunoassay. Have you been able to detect and measure pS129P-alpha synuclein in CSF samples using the IMR immunoassay? Is there evidence in the literature that PD patients commonly have a damaged BBB barrier? Is there any measures of the integrity of BBB barrier (e.g. CSF/serum albumin ratio) available for the patients investigated in the study?
5. (R) Check the following sentence:
Row 55-56; “Post-translational modifications of alpha-synuclein are prone to misfolding”
Author Response
Reviewer 3: In this manuscript the authors investigate if the plasma phosphorylated form of alpha-synuclein is a surrogate marker of PD progression. The measurements of pS129-alpha-synuclein were performed using the ultra-high-sensitivity detection technology for protein immunomagnetic reduction (IMR) immunoassay. I have the following comments:
Question 1:Please provide the anticoagulant used during collection of blood samples.
Reply: We thank the reviewer for the query. Ten milliliters of venous blood was drawn into tubes containing ethylenediamine tetracetic acid (EDTA) as anticoagulants from each participant at enrollment.
This point was added and addressed in the “Experimental Section”, in lines 102-103, on page 3.
Question 2: Clarifications of the statistics used in the manuscripts are needed. Please do not alternate methods for Gausssian distributed variables and non-parametric tests. It would be easier to follow if only non-parametric tests are applied, expressing the values and levels as medians (inter-quartile ranges), Spearman correlations, Mann-Whitney, ANCOVA etc. An alternative, is to provide the statistic method used in each figure, table, results etc. Also providing the evidence for the values that had a Gaussian distribution.
Reply: We thank the reviewer for the kind comments. Because the biomarker data compared in different groups in this study did not follow a Gaussian distribution and violated the assumptions of normality or homoscedasticity, the groups were compared by non-parametric Mann-Whitney U test (for two groups) or the Kruskal-Wallis test (for more than two groups)
We have re-worded these sentences in the “Statistical Analysis” in the “Experimental Section”, in lines 118-121, on page 3. We also briefly addressed the statistical methods again in each table in this revision.
Question 3: There seems be some patients that have extremely high levels of pS129-alpha-synuclein. Have the authors checked if these values could be regarded as ‘outliers’?
Reply: We thank the reviewer for the comments and queries. Yes, we agree with the reviewer that there were 5 outliers with extremely high level of pS129-alpha-synuclein in the old version of Figure 1A. In this revision, the Figure 1A was replaced by a revised new one for the reasons that we added 20 new control samples and we actually have replaced the 5 outliers that the data was beyond 3 standard deviations of the mean value by a repeated double-check data. The data used for the figure 2 and 3 are actually the newly double-checked data of all PD patients. Therefore, the maximal value of pS129- a -synuclein is 67.4 fg/ml. We have detailed the data in the newly added table 3 and amended Figure 1A accordingly. We sincerely thank the reviewer for the kind comment and we apologize for the confusion.
Please refer to the revised Figure 1A on page 5 and Table 3 on page 6.
Question 4: The number of patients with PD (n=122) were many more than the healthy controls (n=48). Was there a certain study design (including the number patients and healthy controls) prior to the analyses of the plasma samples and statistical evaluation of the pS129-alpha-synuclein?
Reply: We thank the reviewer for the comments. We agree with the reviewer that the number of patients with PD were more than the healthy controls. Therefore, we have added additional 20 control samples in this revision with a final total number of 68 controls and 122 PD patients. The sample size in our study reaches significant statistic power (>90%) while considering the mean value of pS129-a-synuclein in each group and setting the type I error as 0.05 (Rosner B. Fundamentals of Biostatistics. 7th ed. Boston, MA: Brooks/Cole; 2011).
This point was listed as one of our study limitations in the “Discussion” section, in line 322 on page 9; and lines 323-324 on page 10.
Question 5: As the authors comment on, we can only speculate regarding the origin of alpha-synclein released into blood. In a recent study by Cariulo et al, Phospho-S129 Alpha-Synuclein Is Present in Human Plasma but Not in Cerebrospinal Fluid as Determined by an Ultrasensitive Immunoassay, they was not able to measure pS129P-alpha-synuclein in CSF samples using an in-house assay based on the Single-molecule counting technology (Singulex Erenna Immunoassay. Have you been able to detect and measure pS129P-alpha synuclein in CSF samples using the IMR immunoassay? Is there evidence in the literature that PD patients commonly have a damaged BBB barrier? Is there any measures of the integrity of BBB barrier (e.g. CSF/serum albumin ratio) available for the patients investigated in the study?
Reply: We thank the reviewer for this important question. However, we don’t have the CSF samples of all participants, including PD patients and controls, in our study. We hence don’t have the chance to examine the phospho-S129 a-synuclein level in the CSF or the measurement of CSF/serum albumin ratio. Indeed, future studies that examine phospho-S129 a-synuclein levels in both CSF and plasma are warranted to further understand the potential role of phospho-S129 a-synuclein as being a biomarker for PD progression.
We have listed this point as one of our study limitations in the “Discussion”, in lines 334-339, on page 10.
Question 6: (R) Check the following sentence: Row 55-56; “Post-translational modifications of alpha-synuclein are prone to misfolding”
Reply: We thank the reviewer for the correction. We have amended this sentence as below to make us clearer.
Post-translational modifications of a-synuclein influence a-synuclein protein prone to misfolding and self-aggregation into fibrillar forms of species.
This point was addressed in the “Introduction”, in lines 54-55, on page 2.
Round 2
Reviewer 2 Report
In the revised version, the authors have addressed all of my objections, and now the manuscript has clearly improved.